# Thoughts of self-harm in adolescents: Relationship with violence in the Dominican Republic

**Kelsey Badger**[1,2]*, **Pamela Baez Caraballo**[3], **Ahzyris Gibbs**[3], **Luz Messina**[3], **Mina Halpern**[3], **Silvia Amesty**[4,5,6]

1 Columbia University Vagelos College of Physicians and Surgeons, New York, New York, United States of America, 2 Department of Psychiatry, Olive View-UCLA Medical Center, Sylmar, California, United States of America, 3 Clínica de Familia, La Romana, Dominican Republic, 4 Center for Family and Community Medicine, Columbia University Irving Medical Center, New York, New York, United States of America, 5 Columbia University Mailman School of Public Health, New York, New York, United States of America, 6 Department of Medical Humanities and Ethics, Columbia University Irving Medical Center, New York, New York, United States of America

* kelseybadger4194@gmail.com

**Data Availability Statement:** The dataset for this article has been uploaded as a Supporting Information file.

## Abstract

Violence against adolescents is a pressing health problem with long-term implications for future physical and mental well-being, such as thoughts of self-harm, which have been associated with suicidal ideation and completion. However, much of the research has been conducted only in high-income countries. This study aimed to examine the correlation between violence against adolescents and self-harm thoughts in La Romana, Dominican Republic (DR). Cross-sectional survey data was collected at a community-based clinic from participants aged 13–20. Participants were recruited through the clinic's adolescent program and peer referral, and verbal consent was obtained. A survey solicited information about each participant's demographics, experiences with violence, and thoughts of harm to self or others. The survey was completed by 49 adolescents. The mean age was 16.78 (SD 2.34); 65% were female. We performed t-tests and Fisher's exact to investigate the relationship between demographics, reported violence experiences and having self-harm thoughts. About half (45%) had experienced physical violence, 76% had experienced emotional violence, and 12% had experienced sexual violence. The most common perpetrators of physical and emotional violence were classmates (12% and 24%), and the most common perpetrator of sexual violence was an ex-partner (4.1%). Ten participants (20.4%) had thought about harming themselves. Self-harm thoughts were significantly associated with being female (p = 0.025), employed (p = 0.05), and to a higher number of experiences of physical (0.029) and sexual violence (p = 0.023). The results of this study suggest a high prevalence of both violence and self-harm thoughts in adolescents in the DR. Interventions that address physical and sexual violence against adolescents may be particularly important. Particular attention should also be paid to screening for self-harm thoughts in female-identifying adolescents. Further research is needed to better understand the relationship between violence and self-harm thoughts in adolescents in the DR.

**Funding:** This project was funded by the Columbia University Vagelos College of Physicians and Surgeons Scholarly Projects program and the Friedman award. (KB) The funders had no role in study design, data collection and analysis, decision to publish, or preparation of the manuscript. None of the authors received salaries from this funding source.

**Competing interests:** The authors have declared that no competing interests exist.

## Introduction

Violence against children is a major public health problem with an estimated one billion children having experienced sexual, physical, or emotional violence in 2022 globally [1]. In Latin America and the Caribbean (LAC), homicide rates for adolescents and young adults (aged 15–29 years) are amongst the highest in the world and the leading cause of death among adolescents [2]. This age group is susceptible not only to physical violence, but also to sexual and dating violence, emotional violence, and bullying [3]. Such susceptibility deepens during middle and late adolescence, where most social and emotional changes occurs [4, 5]. There is a wealth of data supporting that exposure to violence in childhood and adolescence has a long-term negative impacts on mental health. One of first studies to firmly establish the negative effects of exposure to violence in childhood and adolescence on long-term health outcomes was the adverse childhood experiences (ACE) study. This study showed that ACEs, which include childhood abuse and neglect and household dysfunction, are associated with poorer mental health and overall health as adults [6], adding to the literature suggesting that violence experienced during adolescence has been associated to a number of negative mental health outcomes such as risky behavior, substance use, depression, post-traumatic stress, and self-directed violence [7–9]. Specifically, self-harm thoughts have been potentially associated with bullying and violence against adolescents [10], which in turn is associated with increased risk of suicidal ideation and suicide completion [11, 12].

Estimates of the prevalence of self-harm thoughts and behaviors vary widely by country, type of violence event experienced and their association with different mental health symptoms. Overall, it is estimated that up to one fifth of ever-partnered adolescent girls in LAC region have experienced interpersonal violence (IPV) [13]. Lifetime prevalence of self-harm behavior globally in adolescents is approximately 17% [14]. One study from Chile found a 23% prevalence of self-harm among adolescents from low-income backgrounds [15]. One large study conducted in the United States found that emotional maltreatment and neglect in particular were associated with self-harm thoughts [16]. Peer victimization in particular has been linked to self-harm and suicidal ideation [17, 18]. One study that took place in Guatemala showed that in children and adolescents, exposure to violence increased rates of depression and anxiety [19]. Another study conducted in Vietnam showed that emotional violence in particular was associated with poor mental health [20]. Studies have also shown increased rates of suicidal thoughts, depression, and post-traumatic stress in adolescents that have experienced more than one type of violence [21, 22]. One study estimated the rate of bullying in the DR at 44% [23], where bullying was defined as unwanted, aggressive behavior, which involves a real or perceived social power imbalance and is repeated over time [24]. In the Dominican Republic (DR), it has been estimated that around 80% of adolescents have experienced at least one ACE, with the most prevalent being physical abuse and witnessing domestic violence in their household, enforcing its association with depression, anxiety, and dating violence perpetration and victimization [25].

Unfortunately, specific data about self-harm thoughts among LAC countries remains limited. A previous report in the DR estimates that women between the ages of 15–19 years are the most vulnerable to any form of IPV [26]. This calls for more information since there is a knowledge gap for association of violence against adolescents, self-harm thoughts, and the interaction between the two from low- and middle- income countries and from the LAC region. This study is, to our knowledge, the first to examine a relationship between violence against late middle to late adolescents and self-harm thoughts in the DR.

## Methods

### Ethics statement

This study received approval from Columbia University IRB and the Dominican Republic's National Health Bioethics Council (CONABIOS). Verbal assent for adolescents younger than 18, and verbal consent for those 18 and older was obtained from each participant. Adolescents provided assent, and because of the sensitivity of the subjects, parental consent was not obtained. While it is important to ensure the protection of minors, it was thought that requiring parental consent would limit participation, especially those who were most vulnerable [27, 28]. As part of the assent/consent process, participants were assured that their responses to survey at questions would not be shared with anyone outside of the research team.

### Study procedures

This research study took place at a community-based clinic providing healthcare services to children, adolescents, and adults in La Romana, DR. The clinic has an adolescent program that focuses on sexual and reproductive health education, adolescent health services, and family planning. Services are delivered both at the clinic and through outreach at schools and other community centers where youth gather.

Data was collected from August to October 2022. A total of 49 participants aged 13–20 and residents of La Romana, DR were recruited by educators using a two-pronged approach. One part of the recruitment took place during the educators' sessions with adolescents. In that sense, sampling was convenient sampling in nature. To include the adolescents who did not have access to the clinic's educators, we also encouraged adolescents to refer their peers to complete the survey. Participants were not blinded nor randomized due to the small sample and limited resources for the study.

The survey was adapted from surveys the research department at the clinic designed and used in previous studies to estimate incidence of violence in other populations. Furthermore, the survey was pre-tested on small groups of adolescents to ensure clarity and cultural appropriateness, and small changes were made to survey wording and flow to ensure adolescents' understanding of each question posed. This ensured that we kept the most relevant questions to the Dominican adolescent population.

For survey administration, questions were read by a member of the study team to small groups with no more than two to five participants at a time, as we have learned from previous studies done in this community that this is the best approach [29]. Each adolescent filled their own paper survey privately, in order to mitigate response bias. Participants recorded their answers on the paper survey, and the research assistant checked for completeness to avoid missing data, then answers were entered into REDCap, a secure web-based survey [30, 31]. During data collection, authors involved in data collection could have identified individual participants. However, after surveys were entered into REDCap, individual participants were not able to be identified. The survey administration took place in a quiet, private space and took approximately 30 to 40 minutes to complete.

The survey collected socio-demographic information, including age, sex, nationality, level of education, employment status, and composition of household. It also solicited information on experiences with physical, sexual, and emotional/psychological violence, on participants' attitudes about gender and violence, and on help-seeking behaviors and thoughts of harm toward self and others.

Each type of violence—physical, sexual, and emotional/psychological—was defined for participants. They were first asked if they had ever experienced each type of violence. All

participants were asked about experiences regarding specific violent acts: physical violence, was defined as ever been slapped, hit, or kicked; been a victim of attempted kidnapping; been stabbed; been tied up or strangled; or been threatened with a firearm. If participants reported experiencing any of these violent acts, they were asked who perpetrated these acts of violence. Participants were asked about perpetrating violent acts against others, the types of violence committed, and who they had committed the act against. For questions about specific violent acts, perpetrators of violence, and victims of violence, participants were allowed to select more than one of the choices listed.

The survey also included questions about help seeking behaviors. Participants were asked if they currently felt vulnerable to violence, if they had anyone to talk to about experiences with violence, and if they knew where to go if they were a victim of violence. At the end, they were asked if they had ever thought about hurting themselves or someone else, then asked to indicate who they had thought about hurting and if they had thought about seeking help for those thoughts. Psychological support services were available at the clinic for those who were interested.

## Analysis

We began analysis by doing descriptive statistic of each demographic category and violent event reported. Consequently, we explored types of violence experienced or perpetrated by demographic characteristics. We examined whether participants experienced any type of violence perpetrated by others or committed acts of violence. We looked at self-harm thoughts, and types of violence reported by adolescents, those reporting self-harm thoughts and those who did not.

Statistical analysis was performed with R. We began analysis with descriptive statistics of the population, to view distribution by demographic variables, and establish prevalence of suffering by each type of violence. Both the rates of violence by self-report and by rate of having experienced a listed act of violence were reported. For bivariable analysis, we used Fisher's exact test when determining associations between categorical variables, and we used two-sample t-test when comparing two means. A value of $p < 0.05$ was set for statistical significance. The associations were test between having self-harm thoughts, participant demographics and experiencing violence.

## Results

### Demographics

A total of 49 adolescents completed the survey (Table 1). The mean age was 16.78 (SD 2.34). All participants were Dominican, the majority of whom were females (65.3%) from urban areas (95.9%). More than half had not yet to complete high school (55.1%), and one fifth of participants had dropped out of school at some point (18.4%). About 26.5% of participants lived with both parents and sibling(s), and 40.8% lived with one parent. A quarter of the participants were currently employed. None of the participants were married.

### Thoughts of harm

Eighteen participants, 36.7%, had thoughts of causing harm to themselves or others (Table 2), and 20.4% had thought about harming themselves. Almost a quarter of the total sample had thought about seeking help for these thoughts about harming themselves or others.

**Table 1. Sample characteristics (n = 49).**

| Variable | Frequency (%) |
| --- | --- |
| Where participants live | |
| Rural area | 2 (4.1) |
| Urban area | 47 (95.9) |
| Mean age | 16.78 (SD = 2.35) |
| Sex | |
| Male | 16 (32.7) |
| Female | 32 (65.3) |
| Did not know | 1 (2.0) |
| Country of origin | |
| Dominican | 49 (100) |
| Level of education | |
| Not completed high school | 27 (55.1) |
| Completed high school | 22 (44.9) |
| Ever dropped out of school | |
| Yes | 9 (18.4) |
| No | 39 (79.6) |
| No response | 1 (2.0) |
| Lives with | |
| Both parents and siblings | 13 (26.5) |
| One parent | 20 (40.8) |
| Other close family (grandparents, siblings) | 13 (26.5) |
| Other extended family (uncle, step father) | 6 (12.2) |
| Friends | 1 (2.0) |
| Other | 5 (10.2) |
| Currently employed | |
| Yes | 12 (24.5) |
| No | 37 (75.5) |

**Table 2. Thoughts of harm (n = 49).**

| Variable | Frequency (%) |
| --- | --- |
| Ever thought about causing harm to self or others | |
| Yes | 18 (36.7) |
| No | 27 (55.1) |
| No response | 4 (8.2) |
| Thought about harming | |
| Current partner | 1 (2.0) |
| Father | 0 (0.0) |
| Other family member | 4 (8.2) |
| Classmate | 3 (6.1) |
| Self | 10 (20.4) |
| No response | 2 (4.1) |
| Other | 1 (2.0) |
| Ever thought about seeking help for thoughts of harm to self or others | |
| Yes | 12 (24.5) |
| No | 5 (10.2) |
| No response | 1 (2.0) |

### Violence victimization

Almost one third of the participants (30.6%) reported having been a victim of physical violence (Table 3). When asked if they had experienced any of the acts of physical violence listed in the survey, close to half of participants reported having experienced physical violence (44.9%). The most common perpetrator of physical violence was a classmate (12.2%). Almost one in 10 had experienced physical violence at the hands of a stranger (8.2%). When emotional violence was defined, 44.9% of participants reported experiencing emotional violence. However, when asked if they had experienced specific acts of emotional violence ("has anyone ever made you feel embarrassed or humiliated?", "Has anyone ever insulted you?"), three quarters of the sample indicated that they had (75.5%), and many reported experiencing emotional violence perpetrated by a classmate (24.5%), an ex-partner (20.4%), or a family member that was not their father (14.3%). About one in 10 participants reported experiencing sexual violence (12.2%). When asked about specific acts of sexual violence, seven of the 49 participants (14.3%) reported having experienced at least one act of sexual violence. Of the seven participants that experienced sexual violence, three of them (42.9%) experienced violence perpetrated by a partner or ex-partner and three (42.9%) of them chose not to indicate who committed the act against them.

### Violence perpetration

More than one third of participants admitted to having committed an act of physical violence (34.7%; Table 4). Almost one fifth of participants indicated that they had committed an act of physical violence against a classmate (18.4%). More than half of participants reported having committed an act of emotional violence (53.1%) and close to one quarter of participants had committed an act of emotional violence against a classmate (22.5%). Two of the participants (4.1%) reported having committed an act of sexual violence.

### Bivariate analysis

All of the participants reporting self-harm thoughts were female (p = 0.025; Table 5), mean age 18 (p = 0.053). None of the participants reporting self-harm thoughts lived with both parents, compared to one third (33.3%) of those reporting no self-harm thoughts (p = 0.045) who did live with both parents. Half of those reporting self-harm thoughts were currently working compared to only 7.7% of those who reported no self-harm thoughts (p = 0.050).

The mean number of types of violence experienced by those reporting self-harm thoughts was significantly higher than those reporting no self-harm thoughts, 2.20 vs. 1.13 types of violence (p = 0.010). The vast majority (80%) of participants reporting self-harm thoughts had experienced physical violence compared to 35.9% of those reporting no self-harm thoughts (p = 0.029). Half of those reporting self-harm thoughts had experienced sexual violence (p = 0.0023), compared to 5.1% of those reporting no self-harm thoughts.

### Subgroup analysis for female-identifying participants

On our bivariate analysis including only female-identifying participants, we found several factors correlated with having thoughts of self-harm. The mean age of those reporting self-harm thoughts was 18, compared to 16.3 for those reporting no self-harm thoughts (p = 0.053; Table 6). None of the female-identifying participants who reported self-harm thoughts lived with both parents, compared to 36.3% of those who reported no self-harm thoughts (p = 0.035). Half of those reporting self-harm thoughts were working at the time the survey was administered, compared to 9.1% of those reporting no self-harm thoughts (p = 0.019).

**Table 3. Violence victimization (n = 49).**

| Variable | Frequency (%) |
| --- | --- |
| Number of types of violence experienced | |
| 0 | 10 (20.4) |
| 1 | 18 (36.7) |
| 2 | 15 (30.6) |
| 3 | 6 (12.3) |
| Mean number of types of violence experienced | 1.35 |
| Ever experienced physical violence (self-report) | |
| Yes | 15 (30.6) |
| No | 32 (65.4) |
| Did not know | 1 (2.0) |
| No response | 1 (2.0) |
| Ever experienced act of physical violence | |
| Yes | 22 (44.9) |
| No | 27 (55.1) |
| Experienced physical violence perpetrated by | |
| Ex-partner | 3 (6.1) |
| Father | 3 (6.1) |
| Other family member | 3 (6.1) |
| Classmate | 6 (12.2) |
| No response | 1 (2.0) |
| Stranger | 4 (8.2) |
| Other | 3 (6.1) |
| Ever experienced emotional violence (self-report) | |
| Yes | 22 (44.9) |
| No | 25 (51.1) |
| Did not know | 1 (2.0) |
| No response | 1 (2.0) |
| Ever experienced an act of emotional violence | |
| Yes | 37 (75.5) |
| No | 12 (24.5) |
| Perpetrator of emotional violence | |
| Current partner | 3 (6.1) |
| Ex-partner | 10 (20.4) |
| Father | 4 (8.2) |
| Other family member | 7 (14.3) |
| Classmate | 12 (24.5) |
| Teacher | 1 (2.0) |
| No response | 7 (14.3) |
| Stranger | 2 (4.1) |
| Other | 2 (4.1) |
| Ever experienced sexual violence (self-report) | |
| Yes | 6 (12.2) |
| No | 41 (83.7) |
| No response | 2 (4.1) |
| Experienced an act of sexual violence | |
| Yes | 7 (14.3) |
| No | 42 (85.7) |

(*Continued*)

**Table 3.** (Continued)

| Variable | Frequency (%) |
|---|---|
| Perpetrator of sexual violence | |
| Current partner | 1 (2.0) |
| Ex-partner | 2 (4.1) |
| No response | 3 (6.1) |
| Other | 1 (2.0) |

Participants reporting self-harm thoughts had experienced an average number of 2.2 different types of violence compared to those reporting no self-harm thoughts (1.1 different types of violence) (p = 0.0075). Eighty percent of participants reporting self-harm thoughts had experienced physical violence compared to 22.7% of those reporting no self-harm thoughts (p = 0.0051). Half of those reporting self-harm thoughts had experienced sexual violence, compared to 9.1% of those reporting no self-harm thoughts (p = 0.0023).

**Table 4. Violence perpetration (n = 49).**

| Variable | Frequency (%) |
|---|---|
| Committed an act of physical violence | |
| Yes | 17 (34.7) |
| No | 32 (65.3) |
| Committed an act of physical violence against | |
| Ex-partner | 2 (4.1) |
| Other family member | 2 (4.1) |
| Classmate | 9 (18.4) |
| No response | 4 (8.2) |
| Stranger | 1 (2.0) |
| Other | 2 (4.1) |
| Committed an act of emotional violence | |
| Yes | 26 (53.1) |
| No | 23 (46.9) |
| Committed an act of emotional violence against | |
| Current partner | 1 (2.0) |
| Ex-partner | 3 (6.1) |
| Father | 1 (2.0) |
| Other family member | 4 (8.2) |
| Authority figure (police, etc.) | 1 (2.0) |
| Classmate | 11 (22.5) |
| Teacher | 1 (2.0) |
| No response | 4 (8.2) |
| Stranger | 2 (4.1) |
| Other | 3 (6.1) |
| Committed an act of sexual violence | |
| Yes | 2 (4.1) |
| No | 47 (95.9) |
| Committed an act of sexual violence against | |
| Classmate | 1 (2.0) |
| No response | 1 (2.0) |

**Table 5. Bivariate analysis.**

| Variable | Frequency without self-harm thoughts (%) n = 39 | Frequency with self-harm thoughts (%) n = 10 | p-value |
|---|---|---|---|
| Sex | | | |
| Male | 16 (41.0) | 0 (0.0) | *0.025* |
| Female | 22 (56.4) | 10 (100.0) | |
| Did not know | 1 (2.6) | 0 (0.0) | |
| Mean age | 16.46 | 18.00 | 0.053 |
| Level of coursework completed | | | |
| Not completed high school | 22 (56.4) | 5 (50.0) | 0.74 |
| Completed high school | 17 (43.6) | 5 (50.0) | |
| Ever dropped out of school | | | |
| Yes | 7 (7.7) | 2 (20.0) | 1 |
| No | 31 (79.5) | 8 (80.0) | |
| No response | 1 (2.6) | 0 (0.0) | |
| Lives with both parents | 13 (33.3) | 0 (0.0) | *0.045* |
| Does not live with both parents | 26 (66.7) | 10 (100.0) | |
| Currently working | | | |
| Yes | 7 (7.7) | 5 (50.0) | *0.050* |
| No | 32 (82.1) | 5 (50.0) | |
| Ever experienced any violence | | | |
| Yes | 30 (76.9) | 9 (90.0) | 0.66 |
| No | 9 (23.1) | 1 (10.0) | |
| Mean number of types of violence experienced | 1.13 | 2.20 | *0.010* |
| Ever experienced physical violence | | | |
| Yes | 14 (35.9) | 8 (80.0) | *0.029* |
| No | 25 (64.1) | 2 (20.0) | |
| Ever perpetrated physical violence | | | |
| Yes | 14 (35.9) | 3 (30.0) | 1 |
| No | 25 (64.1) | 7 (70.0) | |
| Ever experienced emotional violence | | | |
| Yes | 28 (71.8) | 9 (90.0) | 0.41 |
| No | 11 (28.2) | 1 (10.0) | |
| Ever perpetrated emotional violence | | | |
| Yes | 21 (53.9) | 5 (50.0) | 1 |
| No | 18 (46.2) | 5 (50.0) | |
| Ever experienced sexual violence | | | |
| Yes | 2 (5.1) | 5 (50.0) | *0.0023* |
| No | 37 (94.9) | 5 (50.0) | |
| Ever perpetrated sexual violence | | | |
| Yes | 1 (2.6) | 1 (10.0) | 0.37 |
| No | 38 (97.4) | 9 (90.0) | |

## Discussion

This study aimed to examine the relationship between violence against Dominican adolescents and self-harm thoughts. Our results are consistent with previous studies which have shown higher rates of self-harm thoughts and behaviors in females [14, 32]. With a rate of nearly one third of female-identifying adolescents indicating having thought of self-harm, this indicates a great need for mental health support for this population. Agencies that provide services to this

**Table 6. Bivariate analysis for female-identifying participants only.**

| Variable | Frequency without self-harm thoughts (%) n = 22 | Frequency with self-harm thoughts (%) n = 10 | p-value |
|---|---|---|---|
| Mean age | 16.31 | 18.00 | *0.053* |
| Level of coursework completed | | | |
| Not completed high school | 14 (63.7) | 5 (50.0) | 0.70 |
| Completed high school | 8 (36.3) | 5 (50.0) | |
| Ever dropped out of school | | | |
| Yes | 4 (18.2) | 2 (20.0) | 1 |
| No | 18 (81.8) | 8 (80.0) | |
| Lives with both parents | 8 (36.3) | 0 (0.0) | *0.035* |
| Does not live with both parents | 14 (63.7) | 10 (100.0) | |
| Currently working | | | |
| Yes | 2 (9.1) | 5 (50.0) | *0.019* |
| No | 20 (90.9) | 5 (50.0) | |
| Ever experienced any violence | | | |
| Yes | 16 (72.7) | 9 (90.0) | 0.39 |
| No | 6 (27.3) | 1 (10.0) | |
| Mean number of types of violence experienced | 1.05 | 2.20 | *0.0075* |
| Ever experienced physical violence | | | |
| Yes | 5 (22.7) | 8 (80.0) | *0.0051* |
| No | 17 (77.3) | 2 (20.0) | |
| Ever perpetrated physical violence | | | |
| Yes | 6 (27.3) | 3 (30.0) | 1 |
| No | 16 (72.7) | 7 (70.0) | |
| Ever experienced emotional violence | | | |
| Yes | 16 (72.7) | 9 (90.0) | 0.41 |
| No | 6 (27.3) | 1 (10.0) | |
| Ever perpetrated emotional violence | | | |
| Yes | 13 (59.1) | 5 (50.0) | 1 |
| No | 9 (40.9) | 5 (50.0) | |
| Ever experienced sexual violence | | | |
| Yes | 2 (9.1) | 5 (50.0) | *0.0023* |
| No | 20 (90.9) | 5 (50.0) | |
| Ever perpetrated sexual violence | | | |
| Yes | 0 (0.0) | 1 (10.0) | 0.37 |
| No | 22 (100.0) | 9 (90.0) | |

population should screen for mental health problems, especially self-harm thoughts which could indicate risk of suicide.

None of the males in our study reported having thoughts of self-harm, which could be due to under-reporting. This could be explained by studies showing that those who identify as male tend to under-report mental health issues [33], have a lower tendency to seek help for mental health problems [34, 35], and lower mental health literacy than females [36, 37]. Additionally, it has been reported that men perceive more mental health stigma, particularly among younger men [38]. Future work to reduce the stigma associated with mental illness and increasing awareness could result in higher reporting among male adolescents. In our literature search, no articles addressing mental health stigma in the Dominican Republic could be found. This represents a large gap in knowledge that needs to be filled.

Our results are supported by evidence that sexual violence victimization is associated with high rates of self-harm thoughts and behaviors [39, 40]. This shows the profound impact that experiences with sexual violence can have on the mental health of adolescent girls in the DR. Our study also found that the most common perpetrators of sexual violence were current or former romantic partners, indicating that adolescent dating violence (ADV) may have a significant impact on the mental health of adolescent females, which aligns with studies signaling adolescent women as the most vulnerable group for IPV [26]. ADV is defined as a type of intimate partner violence that can include sexual, physical, or psychological abuse that occurs between two adolescents in a close relationship [41]. Interventions that address ADV could decrease the rates of sexual violence and mental health conditions in Dominican adolescents. Studies have shown that interventions aiming to reduce adolescent dating violence are effective in reducing physical violence and possibly sexual violence among adolescents [42]. These interventions to reduce sexual violence are still a developing area of study, and to our knowledge, there are no studies evaluating these interventions in the DR and very few in Latin America. As such, our study may underestimate the rate of sexual violence for adolescents in the DR because of the stigma associated with sexual violence victimization.

We found that for physical violence, which was also associated with self-harm thoughts, the most common perpetrator was a classmate, despite previous national government initiatives to reduce violence within classrooms [43, 44]. The high prevalence of violence amongst peers adds to a growing body of evidence that suggests the need for increased intervention in schools and other places where adolescents gather aimed at decreasing physical and emotional violence. In addition to physical violence perpetrated by peers, several participants in our study also reported violence perpetrated by family members. Familial abuse has been linked to higher rate of self-harm thoughts in adolescents by several studies [16, 45]. When it comes to physical violence, both peer victimization and familial violence are important factors in self-harm thoughts.

Previous research has shown that adolescents who live in blended households with stepparents have higher rates of self-harm behavior [46]. Non-suicidal self-injury has also been linked to family events such as divorce or death of one parent [47]. This association should be explored further in future studies in order to better understand this potential risk factor.

We found that participants with self-harm thoughts in our study were, on average, older than those without self-harm thoughts. Generally, adolescence is when self-harm behaviors peak, with a decline in prevalence in early adulthood [48, 49]. It is also unclear whether age and employment status were independent risk factors in our study or if because participants with self-harm thoughts were older, they were more likely to be working at the time of the survey. It could also be that the adolescents in our study that were working were from lower-income households where they were required to work. These results align with previous reports proposing that adolescents from low-income backgrounds have higher rates of self-harm [50, 51].

Our study found differences in responses to questions about violence when questions were asked differently. Participants in this study were asked about violence in two ways: asking if adolescents had experienced a type of violence after providing a definition of the violence and after providing a list of acts of violence. More participants answered affirmatively after being provided a list of violent acts. It may be necessary to limit abstraction when answering questions and ask about concrete experiences, particularly when working with adolescents [52]. The difference was especially striking for emotional violence, where almost twice the number of participants responded that they had experienced emotional violence after they were provided with a list of acts. This highlights the need to carefully define types of violence in a concrete way in surveys to avoid underestimating rates of violence in this population.

## Limitations

The most significant limitation in this study is its small sample size. The small sample size limited our options for what types of analysis could be conducted. In addition, since this study was done with a cross-sectional survey, it was susceptible to response and recall bias. Violence tends to be under-reported because experiencing violence is traumatic and stigmatizing, especially for adolescents. The survey may have also been subject to desirability bias, since they were conducted in-person and participants may have been tempted to answer questions in a way that they felt would please the interviewer. To prevent this, however, we encouraged the adolescents to fill out the surveys themselves, and each type of violence was defined by the research assistant prior to survey completion, for the sake of accuracy.

Although previous studies suggests that bullying is highly prevalent in the DR, it is impossible to draw conclusions about these topics from the results of this study. For one, the cross-sectional nature of our study contradicts the definition of bullying, conceptualized as a repeated behavior over time. Additionally, our survey did not have a component that addressed bullying specifically, and while we can infer instances of violence among friends and/or classmates, the responses obtained are not enough to establish causality. In addition, types of relational violence such as spreading rumors or excluding someone from a group, were not measured by the survey. Future studies could explore bullying in this population by asking about repeated behavior and other specific types of violence. Regarding mental health stigma, our study was limited to measuring help-seeking behavior, whether by professional healthcare, or of any kind. It was not within the scope of our project to investigate factors that influence said help-seeking behavior.

Another limitation is that our survey asked only if participants had self-harm thoughts, not if these thoughts were suicidal or not suicidal. Our survey also did not ask if participants had acted on any of these thoughts in the form of non-suicidal self-harm or suicide attempts. Future studies could explore whether participants have engaged in self-harm behavior or had depressive symptomatology in order to better understand risk for suicide and prevalence of non-suicidal self-harm. Furthermore, our recruitment process also limits our results, in that we performed recruitment mainly through the clinic's educators. To reduce this bias, we encouraged adolescents to refer their peers who did not have contact or access to the clinic's educators and invite them to participate in the survey. Even so, our study is limited in the sense that we cannot ensure generalizability, since the experiences of those who were not able willing to participate may be different from the adolescents who did participate in the study.

## Conclusions

More research is needed to understand the patterns of violence against adolescents and self-harm thoughts in adolescents in the DR. Our findings add to a larger body of evidence suggesting a relationship between violence and self-harm thoughts and poor mental health in general. This study is the first, to our knowledge, to attempt to define this relationship for Dominican adolescents, and there is still a need for more research in this field. Sexual violence against adolescents, especially female adolescents, is of particular concern, and more work can be done to prevent and screen for sexual violence in order to identify adolescents at risk for mental health problems. Those that care for youth should consider youth experiences with violence and mental health when evaluating them. This study highlights the need for education about adolescent dating violence, which could help be effective in decreasing sexual violence against adolescents and therefore, promoting mental wellness. Further research is needed to fully characterize self-harm thoughts, behaviors, suicidality, and depressive symptomatology in adolescents from the DR. The high prevalence of peer-to-peer violence in our study highlights

a need to address peer violence and bullying and further examine its effects on the mental health of Dominican adolescents.

## Supporting information

**S1 Checklist. STROBE checklist for observational studies.**
(DOCX)

**S1 Questionnaire. PLOS questionnaire for inclusivity in global research.**
(DOCX)

**S1 Dataset. Study dataset.**
(PDF)

**S2 Dataset. Legend for study dataset.**
(PDF)

## Acknowledgments

We would like to thank the staff at Clínica de Familia La Romana for contributing significant time to this project. This study would not have been possible without their support. We also thank Columbia University for project support and guidance. Above all, we thank the participants in this study for their bravery in sharing their experiences with us.

## Author Contributions

**Conceptualization:** Kelsey Badger, Pamela Baez Caraballo, Mina Halpern, Silvia Amesty.

**Data curation:** Pamela Baez Caraballo, Ahzyris Gibbs.

**Formal analysis:** Kelsey Badger.

**Funding acquisition:** Kelsey Badger.

**Investigation:** Kelsey Badger, Pamela Baez Caraballo, Ahzyris Gibbs, Luz Messina.

**Methodology:** Kelsey Badger, Pamela Baez Caraballo, Luz Messina, Silvia Amesty.

**Project administration:** Pamela Baez Caraballo, Luz Messina, Mina Halpern.

**Resources:** Pamela Baez Caraballo, Luz Messina, Mina Halpern.

**Supervision:** Pamela Baez Caraballo, Silvia Amesty.

**Writing – original draft:** Kelsey Badger.

**Writing – review & editing:** Kelsey Badger, Pamela Baez Caraballo, Ahzyris Gibbs, Mina Halpern, Silvia Amesty.

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
