## [Decision Letter · Decision Letter 0]

3 Jul 2023

PGPH-D-23-00846

Thoughts of self-harm in adolescents: relationship with violence in the Dominican Republic

Dear Dr. Badger,

Thank you for submitting your manuscript to PLOS Global Public Health. After careful consideration, we feel that it has merit but does not fully meet PLOS Global Public Health’s publication criteria as it currently stands. Therefore, we invite you to submit a revised version of the manuscript that addresses the points raised during the review process.

I have carefully reviewed your work and have also received feedback from two reviewers. Both reviewers have noted that minor edits are required, which you will find detailed at the end of this email. While addressing these suggested revisions, I would like to bring to your attention an important concern related to the consent process employed in your study.

Upon reviewing the manuscript, I observed that written informed consent was not mentioned as part of the study procedure. Although verbal consent was obtained from adults aged 18 years and older, verbal assent was sought from participants under the age of 18. Ethically, it is crucial to ensure that the rights and well-being of minors are adequately protected through a comprehensive consent process that includes obtaining written informed consent from their parents or legal guardians. Verbal assent alone may not provide sufficient documentation or assurance of comprehension and voluntary participation.

Therefore, I kindly request that you revise your manuscript to address this concern by incorporating a section on the consent process. Ensure that you obtain written informed consent from the parents or legal guardians of participants under 18 years of age. Additionally, please provide a clear explanation and justification for your chosen consent procedures in the revised manuscript. I notice in the IRB approval letter consent has been partially waived. Please make a mention of it in the manuscript as well.

Furthermore, I have a concern about the data collection tool employed in the study. The development and validity of the questionnaire used in the study needs to be detailed in the methodology section.

In addition to addressing this concern, I also urge you to carefully review and implement the minor edits and suggestions made by me in the attached annotated PDF. By addressing these revisions and adequately addressing the consent issue, you will enhance the ethical rigor and clarity of your study.

We look forward to receiving your revised manuscript.

Kind regards,

Alok Atreya

Academic Editor

Journal Requirements:

2. Our staff editors have determined that your manuscript is likely within the scope of our Global Mental Health: challenges, opportunities, and the future of the field. This editorial initiative is headed by a team of Guest Editors for PLOS GPH: Rochelle Burgess (University College of London) and Dixon Chibanda (University of Zimbabwe and London School of Tropical Medicine and Hygiene). The Collection invites researchers to submit original research which engages with, or disrupts, the urgent needs across the global mental health landscape. We especially encourage submissions of studies that critically interrogate the status quo of the field and that involve inter-/trans-disciplinary approaches and those which share perspectives from underrepresented global regions and communities.

 Additional information can be found on our announcement page: https://collections.plos.org/call-for-papers/global-mental-health-opportunities-challenges/ 

If you would like your manuscript to be considered for this collection, please let us know in your cover letter and we will ensure that your paper is treated as if you were responding to this call.  Please note that being considered for the Collection does not require additional peer review beyond the journal’s standard process and will not delay the publication of your manuscript if it is accepted by PLOS GPH. If you would prefer to remove your manuscript from collection consideration, please specify this in the cover letter

3. Please ensure that Funding Information and Financial Disclosure Statement are matched.

4. In the Funding Information you indicated that no funding was received. Please revise the Funding Information field to reflect funding received.

5. We have noticed that you have uploaded Supporting Information files, but you have not included a list of legends. Please add a full list of legends for your Supporting Information files after the references list. 

6. In the online submission form, you indicated that "Dataset for this article will become available upon reasonable request". All PLOS journals now require all data underlying the findings described in their manuscript to be freely available to other researchers, either 1. In a public repository, 2. Within the manuscript itself, or 3. Uploaded as supplementary information.

Additional Editor Comments (if provided):

Reviewers' comments:

Reviewer's Responses to Questions

**Comments to the Author**

1. Does this manuscript meet PLOS Global Public Health’s publication criteria? Is the manuscript technically sound, and do the data support the conclusions? The manuscript must describe methodologically and ethically rigorous research with conclusions that are appropriately drawn based on the data presented.

Reviewer #1: Yes

Reviewer #2: Yes

2. Has the statistical analysis been performed appropriately and rigorously?

Reviewer #1: I don't know

Reviewer #2: Yes

3. Have the authors made all data underlying the findings in their manuscript fully available (please refer to the Data Availability Statement at the start of the manuscript PDF file)?

Reviewer #1: Yes

Reviewer #2: Yes

4. Is the manuscript presented in an intelligible fashion and written in standard English?

Reviewer #1: Yes

Reviewer #2: Yes

5. Review Comments to the Author

Reviewer #1: My question to author is how would you justify the age of adolescents? According to WHO, 10-19 age groups are considered as adolescent.

Please clarify your sampling technique.

What measures were taken for those having thoughts of harming themselves and others?

Reviewer #2: Line number 137 to 138 please provide the IRC unique ID number and date of the IRC date taken on the bivariate and the sub group analysis please notify that which statistical test was used (t test,chi-square,fisher exact or other )

6. PLOS authors have the option to publish the peer review history of their article (what does this mean?). If published, this will include your full peer review and any attached files.

**Do you want your identity to be public for this peer review?** For information about this choice, including consent withdrawal, please see our Privacy Policy.

Reviewer #1: No

Reviewer #2: No

---

## [Decision Letter · Decision Letter 1]

12 Sep 2023

PGPH-D-23-00846R1

Thoughts of self-harm in adolescents: relationship with violence in the Dominican Republic

Dear Dr. Badger,

Thank you for submitting the revised manuscript to PLOS Global Public Health. After careful consideration, we feel that it has merit but does not fully meet PLOS Global Public Health’s publication criteria as it currently stands. Therefore, we invite you to submit a revised version of the manuscript that addresses the points raised during the review process.

As you know, the original two peer reviewers declined to review this new version. Therefore, I invited a new reviewer, who has now provided comments. While they find the topic valuable, they have raised several important concerns that must be addressed before further consideration:

Major Revisions Needed:

The statistical analyses need to be clarified and potentially revised, as the small sample size may make some tests inappropriate. Please follow the reviewer's statistical advice carefully.More context is needed to situate the findings within the cultural and research landscape of the Dominican Republic. Please expand the introduction and discussion accordingly.The writing requires polish throughout to improve clarity, flow, and grammar. Please thoroughly proofread the manuscript.

In addition to the above major revisions, please also attend to the minor issues raised by the reviewer regarding structure, wording, methods reporting, etc.

I believe addressing these concerns will strengthen the manuscript considerably.

We look forward to receiving your revised manuscript.

Kind regards,

Alok Atreya

Academic Editor

Journal Requirements:

a. State what role the funders took in the study. If the funders had no role in your study, please state: “The funders had no role in study design, data collection and analysis, decision to publish, or preparation of the manuscript.”

b. If any authors received a salary from any of your funders, please state which authors and which funders.

Additional Editor Comments (if provided):

Reviewers' comments:

Reviewer's Responses to Questions

**Comments to the Author**

1. If the authors have adequately addressed your comments raised in a previous round of review and you feel that this manuscript is now acceptable for publication, you may indicate that here to bypass the “Comments to the Author” section, enter your conflict of interest statement in the “Confidential to Editor” section, and submit your "Accept" recommendation.

Reviewer #3: (No Response)

2. Does this manuscript meet PLOS Global Public Health’s publication criteria? Is the manuscript technically sound, and do the data support the conclusions? The manuscript must describe methodologically and ethically rigorous research with conclusions that are appropriately drawn based on the data presented.

Reviewer #3: No

3. Has the statistical analysis been performed appropriately and rigorously?

Reviewer #3: No

4. Have the authors made all data underlying the findings in their manuscript fully available (please refer to the Data Availability Statement at the start of the manuscript PDF file)?

Reviewer #3: Yes

5. Is the manuscript presented in an intelligible fashion and written in standard English?

Reviewer #3: No

6. Review Comments to the Author

Reviewer #3: It is my understanding that this is the second round of revisions. Since I was not a reviewer for the first round, I can not speak to the suggestions being appropriately addressed yet. However, I did find that the statistical analysis is not sound for this particular manuscript.

7. PLOS authors have the option to publish the peer review history of their article (what does this mean?). If published, this will include your full peer review and any attached files.

**Do you want your identity to be public for this peer review?** For information about this choice, including consent withdrawal, please see our Privacy Policy.

Reviewer #3: No

---

## [Editor Report · Decision Letter 2]

22 Nov 2023

Thoughts of self-harm in adolescents: relationship with violence in the Dominican Republic

PGPH-D-23-00846R2

Dear Ms. Badger,

We are pleased to inform you that your manuscript 'Thoughts of self-harm in adolescents: relationship with violence in the Dominican Republic' has been provisionally accepted for publication in PLOS Global Public Health.

Best regards,

Alok Atreya

Academic Editor